# Integrated Secondary Metabolomic and Antioxidant Ability Analysis Reveals the Accumulation Patterns of Metabolites in *Momordica charantia* L. of Different Cultivars

**DOI:** 10.3390/ijms241914495

**Published:** 2023-09-24

**Authors:** Yongxue Zhang, Panling Lu, Haijun Jin, Jiawei Cui, Chen Miao, Lizhong He, Jizhu Yu, Xiaotao Ding, Hongmei Zhang

**Affiliations:** Shanghai Key Laboratory of Protected Horticulture Technology, Horticultural Research Institute, Shanghai Academy of Agricultural Science, Shanghai 201403, China; xuezylemon@foxmail.com (Y.Z.); lpl2245@163.com (P.L.); jinhaijun@saas.sh.cn (H.J.); cuijiawei@saas.sh.cn (J.C.); miaochen@saas.sh.cn (C.M.); hlznd02@163.com (L.H.); yy2@saas.sh.cn (J.Y.)

**Keywords:** antioxidant activity, bitter gourd, flavonoids, metabolites, phenolic acids

## Abstract

Bitter gourd (*Momordica charantia* L.) contains rich bioactive ingredients and secondary metabolites; hence, it has been used as medicine and food product. This study systematically quantified the nutrient contents, the total content of phenolic acids (TPC), flavonoids (TFC), and triterpenoids (TTC) in seven different cultivars of bitter gourd. This study also estimated the organic acid content and antioxidative capacity of different cultivars of bitter gourd. Although the TPC, TFC, TTC, organic acid content, and antioxidative activity differed significantly among different cultivars of bitter gourd, significant correlations were also observed in the obtained data. In the metabolomics analysis, 370 secondary metabolites were identified in seven cultivars of bitter gourd; flavonoids and phenolic acids were significantly more. Differentially accumulated metabolites identified in this study were mainly associated with secondary metabolic pathways, including pathways of flavonoid, flavonol, isoflavonoid, flavone, folate, and phenylpropanoid biosyntheses. A number of metabolites (*n* = 27) were significantly correlated (positive or negative) with antioxidative capacity (r ≥ 0.7 and *p* < 0.05). The outcomes suggest that bitter gourd contains a plethora of bioactive compounds; hence, bitter gourd may potentially be applied in developing novel molecules of medicinal importance.

## 1. Introduction

Throughout history, numerous plant species have been utilized for medicinal and culinary purposes across the globe. One member of the Cucurbitaceae family, *Momordica charantia* L. (a vine), is an example [1]. Due to its distinctively bitter flavor, there are several common names for this plant species, such as bitter melon, bitter gourd, balsam pear, kugua, or karela [2]. The various cultivars of bitter gourd are extensively grown in tropical and subtropical regions worldwide [3]. During harvesting, the pericarp color of bitter gourd can be dark green, pale green, or white [4]. In general, immature bitter melons with dark green skin have a stronger bitter taste than those with lighter skin color because light green bitter gourds have a lower content of saponins [5,6]. In addition, at maturity, the bitter taste of bitter gourds is relatively low, which is possibly due to their high carotenoid content [7]. The fruit of this plant contains vitamins, minerals, and flavonoids, and these fruits are utilizable as food at each step till they mature; this plant species is a common vegetable globally [4]. Furthermore, in a few countries, different parts such as fruits, seeds, stems, and leaves of bitter gourd have long been implemented as medicines for the treatment of diseases, such as intestinal diseases. These parts also contain anti-tumor and anti-aging properties and were used to maintain blood sugar, blood pressure, and cholesterol [8]. Phytochemicals, such as peptides, proteins, phenolic compounds, saponins, triterpenes steroids, alkaloids, and polysaccharides, that are found in bitter gourd are mainly responsible for its health-promoting properties [1,9,10].

Primary metabolites play crucial functions during the growth and development of plants; furthermore, these compounds are highly conserved in plants [11]. The lineage of secondary metabolites is often specific, and these metabolites govern the plants’ interaction with biotic and abiotic environments [12]. Plants synthesize several diverse types of secondary metabolites, including different phytochemicals and natural products [13]. Many secondary metabolites, such as sterols, phenolic acids, saponins, flavonoids, and alkaloids, are natural medicines and can be utilized for the treatment of different diseases [14]. Historically, natural products produced by plants have been studied in different drug discovery projects [15]. Recently, untargeted metabolomic approaches have been used to investigate bitter gourd’s dynamic metabolite alterations in response to abiotic stress [16,17]. Notably, more than 90 flavonoids have been identified in bitter gourd [4]; however, studies on several other secondary metabolites are lacking. Bitter gourds have been used for a long time; however, systematic studies on the peel color and active ingredients of bitter gourds from different regions are lacking. Studies comparing different varieties of bitter gourd are also rare. In terms of methods employed for detecting metabolites in bitter gourd, previous studies have only focused on qualitative and quantitative analyses of flavonoid metabolites [4], with no studies on other secondary metabolites. Therefore, non-targeted qualitative analysis techniques are required to identify chemical components of bitter gourd (on a large scale) and maximize the utilization of different varieties of bitter gourd plants. The evaluation of the correlation of antioxidant capacity with metabolite accumulation will reveal the presence of antioxidative and chemical components in bitter gourd plants and provide a reference for improving the utilization of commercial bitter gourd varieties.

Using physiological and biochemical experiments, this study determined the contents of nutrients, phenolic acids, flavonoids, triterpenoids, and organic acids in different cultivars of bitter gourd. The antioxidative activity of these compounds was also investigated. Furthermore, ultra-performance liquid chromatography–tandem mass spectrometry was used to qualitatively and quantitatively examine the secondary metabolites of seven distinct bitter gourd samples. This study showed that different varieties of bitter gourd accumulate diverse types of metabolites. The results increase our understanding of different chemical compounds found in plants belonging to the Cucurbitaceae family, allowing the development of antioxidants and resources of medical importance using bitter gourd plants.

## 2. Results

### 2.1. Comparison of Characters and Nutrients of Different Cultivars

Different bitter gourd cultivars demonstrated diversified characters (Figure 1). The fruits of seven cultivars of bitter gourd were divided into the following types based on the fruit skin color: white (PG and CB), white-green (BFM), green (LJ and LBS), and dark green (JLZ and RB). Furthermore, fruits could be divided into the following types based on their shape: circular (PG), fusiform (JLZ, CB, and LJ), and long conical shapes (BFM, RB, and LBS). Notably, the shape of the melon tumor was very different among these cultivars. For instance, PG and BFM had pearl-shaped tumors, JLZ and RB had dense strip tumors, CB and LJ contained long-grain tumors, and LBS had thick strip tumors.

The nutrient contents differed significantly among seven bitter gourd cultivars (Table 1). The content of vitamin C (Vc) in PG, JLZ, RB, and CB was higher (>114 μg g^−1^), whereas the content of Vc in BFM, LJ, and LBS was similar (56.65–65.28 μg g^−1^). The soluble protein content of seven bitter gourd cultivars ranged from 4.70 to 13.30 mg g^−1^ (average, 8.51 mg g^−1^). JLZ and LBS have the highest soluble protein content; 2.8 and 2.7 times and 2.6 and 2.5 times higher than PG and BFM, respectively. The amino acid content of different cultivars ranged from 0.50 to 2.03 mg g^−1^ with significant differences. Compared with RB, the amino acid content of BFM and LBS increased four times. The cellulose content of different bitter gourd cultivars ranged from 274.94 mg g^−1^ to 399.46 mg g^−1^ (average, 319.61 mg g^−1^). The cellulose content in BFM and CB was the highest, whereas JLZ, RB, and LBS had the lowest cellulose content. The total acid content did not differ significantly among different cultivars (mean, 0.41%; range 0.28% to 0.52%).

### 2.2. Organic Acid Content

Four organic acids, namely, oxalic acid, succinic acid, malic acid, and citric acid, were detected in all bitter gourd cultivars. Although malic acid was predominant in all studied bitter gourd cultivars, significant differences were observed in terms of the average content of organic acid among different cultivars (Table 2). The spindle-shaped JLZ and CB were found to contain the largest amounts of organic acids. In terms of individual acids, LJ and LBS had the highest and lowest oxalic acid content, respectively. On the other hand, CB and LBS had the highest and lowest succinic acid content, respectively. The JLZ and CB cultivars had the highest content of malic acid, whereas the lowest malic acid content was observed in the LBS cultivar. The content of citric acid in different cultivars followed a trend identical to that of malic acid.

### 2.3. Total Phenolic, Flavonoid, and Triterpenoids Contents

Table 3 shows the total phenolic content (TPC), total flavonoid content (TFC), and total triterpenoids content (TTC) of different cultivars of bitter gourd. The TPC of the LJ cultivar was 1.71 times higher than that of PG and JLZ, whereas the TFC of the LJ cultivar was 1.4 times higher compared with that of the PG cultivar (Table 3). The TPC content ranged from 1.23 mg g^−1^ (PG) to 1.72 mg g^−1^ (CB), and the TPC content in the CB cultivar was 1.4 times higher than those of PG and JLZ cultivars (Table 3). CB, LJ, and LBS had higher TPC, TFC, and TTC, whereas these compounds were lower in PG and JLZ. Cultivars representing high TPC, TFC, and TTC might be more suitable to act as abundant sources of natural phenolic compounds. It was also found that the LJ cultivar had higher TPC and TFC than those of the other six cultivars; hence, it can be speculated that this cultivar has high antioxidative activity. Thus, the nutritional and health effects of different cultivars of bitter gourd may be different. The changes of TPC, TFC, and TTC in the pulp of bitter gourd were greatly influenced by regional and variety-related factors.

### 2.4. Antioxidant Activity

This study investigated the anti-oxidative activity of different cultivars of bitter gourd using several assays. Since α, α-diphenyl-β-picrylhydrazyl (DPPH) is among the most stable free radicals, this molecule is widely employed to understand the presence of free radical scavengers in natural foods. This study demonstrated that the average antioxidative activity (expressed as percent inhibition) of seven cultivars of bitter gourd ranged from 0.84% to 0.58% (Table 3). Extract prepared from the CB cultivar displayed the most potent DPPH scavenging activity; this was 1.45 times higher than that of the lowest activity displayed in the PB extract. Extracts from LJ and LBS cultivars showed 1.3 times higher DPPH scavenging activity compared with the lowest activity value. When 2,2-azino-bis-3-ethylbenzothiazoline-6-sulphonic acid (ABTS) scavenging activity was measured, different cultivars showed a diverse range of antioxidative activity (Table 3). The highest and lowest ABTS scavenging activities were demonstrated by extracts of LJ (13.00 μmol g^−1^ FW) and PG (10.11 μmol g^−1^ FW) cultivars, respectively. Notably, a similar trend of antioxidative activity was observed among different cultivars in both ABTS and DPPH radical scavenging assays. When ferric ion reducing antioxidant power (FRAP) was employed to measure the antioxidative activity of different cultivars, the LJ extract showed the strongest reducing antioxidant power (13.93 μmol g^−1^ FW), which was at least 1.5 times higher than that of the lowest activity (9.64 ± 0.62 μmol g^−1^ FW; the PG extract) (Table 3). Our study revealed similar trends in the antioxidative activity assays and TPC, TFC, and TTC in different cultivars of bitter gourd, suggesting that phenolic acids and flavonoids are crucial for these cultivars in terms of their antioxidative capacity.

### 2.5. Correlation of TPC, TFC, and TTC with Antioxidative Capacity

A significant correlation of TPC, TFC, and TTC with antioxidative capacity was observed (*p* < 0.05) in a few cultivars of bitter gourd (Table 4). For instance, PG and JLZ showed a significant correlation of TPC, TFC, and TTC with antioxidative capacity; however, no such correlation was observed in the case of BFM, LJ, and LBS cultivars. For the PG cultivar, the TPC correlation coefficients were −0.983 and 1 for ABTS and FRAP, respectively; furthermore, the TFC correlation coefficients for ABTS and FRAP were 0.995 and −0.995, respectively, whereas the TTC correlation coefficients for ABTS and FRAP were 0.994 and −0.996, respectively. Hence, TPC had significant positive correlations with FRAP, whereas TFC and TTC exhibited significant positive correlations with ABTS. This study also revealed a significant relationship of TPC with TFC as well as TTC, and the correlation coefficient for both relationships was −0.977 (Table 4). For the JLZ cultivar, the TPC correlation coefficients were −0.925, −0.991, and −0.976 for DPPH, ABTS, and FRAP, respectively, but in this cultivar, TFC (all the coefficients were above 0.972) was inversely correlated with TPC (Table 4). In the case of LJ and LBS cultivars, no significant correlation of TPC, TFC, and TTC was observed with antioxidative indices (Table 4). This could be explained by the fact that, in these cultivars, other nonphenolic compounds also contribute to antioxidative activity. Consistent with previous studies, the fruit soluble solids, titratable acids, total phenolic contents, antioxidative capacity, and antioxidative enzyme activities varied with genotypes (‘B-76’, ‘B-59’ and ‘SHF-3A’) of *Vaccinium stamineum* L. [18].

### 2.6. Profiles of the Secondary Metabolome in Seven Cultivars of Bitter Gourd

In the overlapping analysis of total ion current (TIC) in different quality control (QC) samples, the results showed that UPLC-MS had good instrumental stability, and it significantly improved the reproducibility and reliability of metabolite extraction and detection (Appendix A). Furthermore, Pearson’s correlation analysis demonstrated strong correlations within different replicate samples (Appendix A).

Three-dimensional principal component analysis (PCA) was used to verify the differences among seven tested cultivars of bitter gourd (Figure 2A). Except for that of the LBS cultivar, the profiles of the other six cultivars showed separation. In addition, all three biological replicates of the same cultivar were closely related, indicating stable and repeatable metabolomics data. Figure 2B shows a heatmap, developed based on HCA, which exhibited accumulated patterns of all identified metabolites for all the seven cultivars of bitter gourd. This analysis revealed the similarity of components among biological repeats and the differences of components among different cultivars of bitter gourd. In summary, the results of the HCA were consistent with those of the PCA results. Among the seven cultivars of bitter gourd, the metabolite profiles could be divided into the following three categories: (1) JLZ and BFM, (2) CB and LJ, and (3) PG, LBS, and RB.

The metabolomics approach led to the identification of a total of 370 secondary metabolites; these included phenolic acids, flavones, flavanones, flavanols, isoflavones, alkaloids, organic acids, terpenoids, lignans, anthocyanins, coumarins, quinones, and others (Appendix A). Among them, phenolic acids (76, 20.54%), flavones (72, 19.64%), alkaloids (56, 15.14%), and organic acids (50, 13.51%) accounted for a substantial proportion of all metabolites (Figure 2C). Since flavonoids and phenolic acids are the major antioxidants in plants [19], they were also found to constitute the largest proportion of bitter gourd (Figure 2C).

### 2.7. Differential Metabolite Analysis of Seven Cultivars of Bitter Gourd

Different patterns of metabolite accumulation were observed when the metabolic analysis of all samples was conducted. The number of up-regulated and down-regulated compounds was analyzed by pairwise comparisons. Since JLZ is a wild variety [20], it was used as the control in this study. To identify differentially accumulated metabolites (DAMs), 370 annotated metabolites were screened based on the following criteria: fold change (FC) ≥2 or ≤0.5 and variable importance in projection (VIP) value of ≥1; metabolites satisfying these criteria were then considered as DAMs (Appendix A).

A pairwise comparison of the examined bitter gourd samples was used to determine the number of chemicals that were elevated and down-regulated (when JLZ was considered as the control). There were 138 DAMs (32 up-regulated, 106 down-regulated) between PG and JLZ, 70 DAMs (26 up-regulated, 44 down-regulated) between BFM and JLZ, 74 DAMs (36 up-regulated, 38 down-regulated) between RB and JLZ, 73 DAMs (54 up-regulated, 19 down-regulated) between CB and JLZ, 70 DAMs (32 up-regulated, 38 down-regulated) between LJ and JLZ, and 79 DAMs (26 up-regulated, 53 down-regulated) between LBS and JLZ. Three compounds were commonly differentiated among the six comparison groups. These three compounds were methyl gallate (phenolic acid), persicoside (flavanones), and pyrrole-2-carboxylic acid (others); methyl gallate and persicoside were upregulated in six treatment groups, whereas pyrrole-2-carboxylic acid was increased in the five treatment groups except the PG and JLZ treatment groups (Figure 3B).

When the DAMs were functionally annotated using the Kyoto encyclopedia of genes and genomes (KEGG) database, it was found that the DAMs for PG vs. JLZ, BFM vs. JLZ, RB vs. JLZ, CB vs. JLZ, LJ vs. JLZ, and LBS vs. JLZ were involved in 39, 21, 26, 34, 26 and 29 pathways, respectively (Appendix A). Figure 3C shows the top 20 most enriched differential-metabolite-associated pathways. Each group also had unique metabolic pathways. Pathways of biosynthesis of phenylalanine, tyrosine, and tryptophan, D-arginine and D-ornithine metabolism, and the sulfur relay system were unique in PG vs. LJZ; folate biosynthesis was unique in BFM vs. LJZ; cysteine and methionine metabolism was unique in CB vs. LJZ; glycerolipid metabolism, stilbenoid, diarylheptanoid, and gingerol biosynthesis, and the pentose phosphate pathway were unique in LJ vs. LJZ; arginine biosynthesis, histidine metabolism, C5-branched dibasic acid metabolism, pentose, and glucuronate interconversions were unique in LBS vs. LJZ (Figure 3C, Appendix A). Meanwhile, flavonoids, flavones and flavonols, isoflavonoids, and phenylpropanoid biosynthesis pathways were enriched. These results could be the reason for higher TPC and TFC (along with strong antioxidative activities) in LJ and LBS cultivars (Figure 3C).

### 2.8. Network Diagram between Antioxidative Capacity and Metabolites

We analyzed the Pearson correlation coefficient between the numerous metabolites detected and the antioxidant capacity assays (DPPH, ABTS, and FRAP) to understand further the antioxidants in bitter gourd. Consequently, using the positive or negative associations (r ≥ 0.7 and *p* < 0.05) between antioxidative ability, 27 different metabolites were observed (Figure 4). Of these metabolites, 15 phenolic acids (1-*O*-galloyl-β-d-glucose, 3,4,5-trimethoxyphenyl-β-d-glucopyranoside, 3,4-dihydroxybenzeneacetic acid, 4-hydroxybenzaldehyde, 4-*O*-(6′-*O*-glucosyl-4”-hydroxybenzoyl)-4-hydroxybenzyl alcohol, chlorogenic acid methyl ester, caffeic acid, coniferyl alcohol, crypto chlorogenic acid, feruloyl glucose, homogentisic acid, hydroxy-methoxycinnamate, malonyl-caffeoylquinic acid, p-hydroxyphenyl acetic acid, and sinapic acid), five other acids (benzamide, biotin, mevalonic acid, orychophramarin A [7,8-dihydroxy-3-(4-hydroxyphenyl)-isocoumarin], and phenethylamine), 3 organic acids (2-methyl glutaric acid, 4-guanidino butyric acid, and -ketoglutaric acid), 2 flavones (chrysoeriol 6-C-hexoside 8-C-hexoside-*O*-hexoside and mearnsetin-3-*O*-glucoside), 1 flavanone (persicoside), and 1 lignin (syringaresinol-hex) were significantly positively or negatively correlated with at least one antioxidative capacity index. This suggested that bitter gourd contains a variety of antioxidants, not only phenolic acids and flavonoids (Figure 4, Appendix A).

## 3. Discussion

This study investigated the metabolic constituents in seven different cultivars of bitter gourd and found many notable changes and differences in several metabolites among these cultivars. The metabolic details identified may allow the development of products of medical importance from bitter gourd.

### 3.1. Comparison of Antioxidant Activity in Different Cultivars of Bitter Gourd

Bitter gourd has been shown to be an effective antioxidant both in vitro and in vivo, making it a good candidate for usage as a dietary supplement [21,22]. According to our findings, different cultivars of bitter gourd had different TPC, TFC, TTC, and total antioxidant activities (Table 3). Similar to this, earlier findings demonstrated that various mulberry cultivars were gathered from the same growth environment, but that there were significant variations in TPC, TFC, and TTC content, which may be influenced by genetic variation [23]. However, among various cultivars of bitter gourd, there varied degrees of correlation between antioxidant activity and the content of each substance (Table 4). This study discovered that the TPC of four bitter gourd cultivars had a strong correlation in the FRAP assay (Table 4). Previous research has also found a correlation between TPC and FRAP in bitter gourds [24], consistent to the results of our study. There is a correlation between various metabolites and antioxidant capacity in addition to the content of each substance. This association takes into account metabolites of phenolic acids and flavonoids as well as certain alkaloids and other potentially powerful antioxidants found in bitter gourd (Figure 4, Appendix A). Similarly, the results of the correlation among different metabolites and antioxidantive capacity in 22 *Lilium* spp. They also revealed that in addition to phenolic acids and flavonoids, a number of alkaloids, quinones, and terpenoids are also related to antioxidant capacity, and these compounds may also be important antioxidants [25].

### 3.2. Phenolic Acids and Flavonoid Components

Phenolic and flavonoid compounds are important components of bitter gourd [26]. This study identified 56 phenolic acid metabolites in seven cultivars of bitter gourd. These included (a) metabolites associated with phenylpropanoid biosynthesis pathway (caffeic acid, chlorogenic acid, coniferaldehyde, coniferin, coniferyl alcohol, ferulic acid, p-coumaryl alcohol, sinapic acid, sinapoyl malate, and syringin); through this pathway, plants can synthesize many important compounds, such as flavonoids, coumarin, and lignin, (b) that affect the accumulation of carbohydrates and the levels of a few organic acids and derivatives (such as chlorogenic acid, di-*O*-glucose-quinic acid, sinapoylcaffeoyltartaric acid, and methyl gallate), and (c) metabolites triggering the biosynthesis of a few chelators (hydroxy-methoxycinnamate and protocatechuic acid-4-glucoside) (Figure 3C, Appendix A). This study showed that the metabolic pathways enriched with the KEGG pathway in bitter gourd were mostly related to the biosynthesis of flavonoids and phenylpropanoid (Figure 3C). Phenylalanine is a precursor for the biosynthesis of para coumaric acid, and the products of para coumaric acid are caffeic acid and ferulic acid [27]. Previously, the primary phenolic acids in bitter gourd flesh were isolated using HPLC and were found to be gallic acid, chlorogenic acid, catechin, gentisic acid, and epicatechin [27,28]. Catechin, which accounts for 72–86% of bitter gourd extracts’ total phenolic contents, is the most abundant phenolic acid in bitter gourds, followed by gentisic acid (4–12%), gallic acid (0.25–0.87%), and chlorogenic acid (0.02–0.26%), as reported by Budrat and Shotipruk [29]. Another study showed that the content of caffeic acid in the methanol extract of bitter gourd was 3.55 mg L^−1^ [24]. Bitter gourd aqueous extract fractions have also been shown to contain p-coumaric acid, tannic acid, benzoic acid, ferulic acid, gallic acid, caffeic acid, and (+)-catechin [30]. The distribution of phenolic acid content varied in different bitter gourd tissues [27]. There was a wide variation in the phenolic content of the bitter gourd, with the seed containing 4.67 to 8.02 mg chlorogenic acid equivalent (CAE) g^−1^ DW, the inner tissue containing 4.64 to 8.94 mg CAE g^−1^ DW, and the flesh containing 5.36 to 8.90 mg CAE g^−1^ DW [27]. Further, it was discovered that the bitter gourd’s flesh has a significantly higher phenolic acid concentration than the inner tissues and seeds [27]. Phenolic compounds have antioxidant, insecticidal, antiparasitic, anti-inflammatory, antibacterial, anti-diabetic, wound healing, antidiuretic, cytotoxic, and antitumor activities [31,32]. Further studies on the isolation of unknown active principles from bitter gourd may bring out a new source for anti-inflammatory and anti-cancer drugs.

Meanwhile, flavonoids, as secondary metabolites of plants, play essential roles in many biological processes, and most flavonoids are synthesized downstream of phenolic acids [25,33]. In the present study, we identified 42 flavonols, 21 flavones, 4 isoflavones, and 2 flavanones in seven cultivars of bitter gourd (Appendix A). Meanwhile, we identified multiple metabolites involved in the flavone and flavonol biosynthesis, including apigenin 7-*O*-glucoside (Cosmosiin), kaempferol 3-*O*-galactoside (Trifolin), kaempferol-3-*O*-glucoside (Astragalin), kaempferol 3-*O*-rhamnoside (Kaempferin), kaempferol-3-*O*-rutinoside (Nicotiflorin), luteolin 7-*O*-neohesperidoside (Lonicerin), quercetin 3-*O*-a-L-rhamnoside (Quercitrin), quercetin 3-*O*-glucoside (Isotrifoliin), luteolin-7-*O*-glucoside (Cynaroside), quercetin-3-*O*-methyl ether, and quercetin-3-*O*-rutinoside (Rutin) (Appendix A). The expression of most flavonoids in the PG cultivar with a white pericarp was down-regulated when compared with that of the JLZ cultivar with a dark green pericarp. Notably, however, CB and PG cultivars with white pericarp did not show identical expression patterns of flavonoids (Appendix A). The relative flavonoid content in bitter gourds of different colors was also found to be different [4]. Among other flavonoid-type compounds, flavones and flavonols are distinct from colored anthocyanins because these compounds are colorless [34]. Furthermore, these colorless compounds are found in only certain types of cells, for instance, the epidermal cells of flowers. Accumulation of flavonols was observed in all plant organs; however, these compounds were unevenly distributed in different plant cells [35]. Flavones and flavonols may have a great influence on the colors of chrysanthemum flowers [35]. Hence, flavonoid compounds are also important for the color, quality, and commercial value of plants. The pharmacological effects of these substances are extensive, ranging from anti-inflammatory to antioxidant, with additional benefits including regulation of autophagy and amelioration of mitochondrial dysfunction [36,37,38,39,40,41,42,43]. For example, apigenin 7-*O*-rutinoside reduces the risk of reactive oxygen species-related conditions [44]. Genistein is an isoflavone that has estrogen-like functions in mammals [45]. Meanwhile, genistein has antioxidant, anti-inflammatory, antibacterial and antiviral activities, angiogenic and estrogenic effects, and pharmacological activities on diabetes and lipid metabolism [46]. We found that in PG and CB cultivars with white pericarp, apigenin 7-*O*-rutinoside and genistein were highly expressed, respectively (Appendix A). Our results suggest that depending on the pericarp color, bitter gourd may show differing medicinal capability, and this difference emerges from the secondary metabolites’ content. Hence, variations in the secondary metabolite composition may provide different kinds of bitter gourd with varied therapeutic purposes.

### 3.3. Terpenoids Components

Terpenoids are the most diverse chemicals produced by plants. Most of the terpenoids produced by plants protect the growth of plants in abiotic and biotic conditions by carrying-out specific interactions [47,48]. Tetriterpenes and their glycosides are the main chemical components of saponins. Most of these compounds have a bitter taste and show toxicity; hence, they are termed cucurbitanes [49]. The antidiabetic and hypoglycemic activities of bitter gourd were attributed to saponins or cucurbitanes [50]. We identified several terpenoids in seven different cultivars of bitter gourd and demonstrated that many of these terpenoids were down-regulated compared with the levels in the JLZ cultivar (Appendix A). The methanol extract of bitter gourd showed the presence of lucyoside (F, H, M, O, and Q types) (Appendix A). These compounds are organic compounds known as triterpene saponins and can regulate the host’s immune function [51]. Cucurbitacin B and E are widely distributed in Cucurbitaceae with important biological and pharmacological activities [52]. Cucurbitacin E (CuE) has anti-inflammatory and anti-tumorigenic properties, which are related to its effect on the cellular cytoskeleton, mitotic pathways, and cellular autophagy [53]. A study recently demonstrated that CuB has effective anti-tumor activities in lung cancer [54]. Several different cucurbitane-type compounds, such as cucurbitacin B *O*-rhamnoside, cucurbitacin D-*O*-glucoside, cucurbitacin E *O*-rhamnoside, and cucurbitacin E *O*-glucoside, have been identified from bitter gourd fruits. These compounds may be responsible for the inhibitory activity of bitter gourd against testosterone-induced prostatic hyperplasia in mice [55]. We also identified several cucurbitane-type triterpenoids (e.g., 23,24-dihydro cucurbitacin E, dihydroisocucurbitacin I-glucoside, 23,24-dihydro cucurbitacin D-*O*-glucoside, and dihydroisocucurbitacin I-malGlu) from seven cultivars of bitter gourd in the present study. Although triterpenoid saponins showed cytotoxic and antivirus activities [56], further isolation and identification of active terpenoid saponins from bitter gourd species are needed.

### 3.4. Organic Acids Components

Organic acids are a crucial part of the composition of the entire fruit, greatly influencing the flavor, texture, freshness, and nutritional and medicinal values of food [57]. This study detected several organic acids in bitter gourd, and their contents depended on different cultivars. Malic and oxalic acids were the two major organic acids identified in the bitter gourd fruit (Table 2). In a previous study, six main orgainc acid were identified in pomegranate (*Punica granatum* L.), with malic acid, citric acid, succinic acid, and oxalic acid accounting for 97% of the total acid content. Furthermore, tartaric acid and ascorbic acid were less abundant. Interestingly, malic acid was the main organic acid in Tunisian pomegranates and the cultivar ‘Mezzi 2’. Citric acid has been reported as a prominent content of Turkish pomegranates, whereas oxalic acid was the main organic acid in the ‘Assaria’ Portugal cultivar and in 40 Spanish cultivars [58]. In cells, organic acids are mainly produced by the (1) tricarboxylic acid cycle and glycolytic pathway and (2) direct glucose oxidation [59]. Furthermore, we identified 23 metabolites of organic acid in seven cultivars of bitter gourd (Appendix A). For instance, jasmonic acid (JA) and N-[(−)-jasmonoyl]-(L)-isoleucine (JA-L-Ile) were up-regulated in bitter gourd (Appendix A). JA, which is an endogenous growth-regulating compound, is synthesized by higher plants mainly using the octadecane (starts from α-linolenic acid [18:3]) and hexadecane (starts from hexadecane trienoic acid [16:3]) [60]. Subsequently, (13S)-12,13-epoxy-octadecatrienoic acid (12,13-EOT) and (11S)-10,11-EOT are formed, which are rapidly converted into cis-12-oxo-phytodienoic acid (OPDA) and dinor-oxo-phytodienoic acid (dnOPDA) in the chloroplast. Later, OPDA and dnOPDA were transferred from plastid to peroxisome and reduced to 3-oxo-2-(2-(Z)-pentenyl)-cyclopentane-1-octanoic (OPC-8) and OPC-1-hexanoic (OPC-6). OPC-8 and OPC-6 then undergo several rounds of β-oxidation to form OPC-1-butanoic acid (OPC-4) and finally JA in the peroxisome. In the cytoplasm, JA is metabolized into different structures by various chemical reactions, such as MeJA, JA-Ile, cis-jasmone (CJ), and 12-hydroxyjasmonic acid (12-OH-JA) [61]. JAs are involved in the regulation of important plant growth and development, and especially the mediation of plant responses to biotic and abiotic stresses, which can inhibit Rubisco biosynthesis, induce stomatal opening and affect the uptake of nitrogen and phosphorus and the transport of organic matter such as glucose [62]. Suberic acid is abundant in several plants, including Vernonia galamensis, castor, and Hibiscus syriacus [63]. We also identified an increase in suberic acid in bitter gourd (Appendix A). The molecular function of suberic acid remains unknown; however, it can protect hairless mice from ultraviolet light-mediated skin aging by increasing collagen content and collagen synthesis genes (COL1A1) [64]. Methylmalonic acid (MMA), which is also identified in the present study, is a marker for detecting subclinical vitamin B12 deficiency [65]. Meanwhile, several organic acids involved in tricarboxylic acid cycle metabolism were also identified in bitter gourds. These organic acids are involved in cellular metabolism and energy production [66,67]. Because organic acids can reduce the pH of food and hence show bacteriostatic activities, they are also used as additives and preservatives [59].

### 3.5. Other Components

This study identified 27 alkaloids metabolites in seven cultivars of bitter gourd; these metabolites were associated with different metabolic pathways (Appendix A). With tyrosine, tryptophan, ornithine, and lysine acting as precursors, tetrahydroisoquinoline, indole, pyrrolizidine, and piperidine alkaloids, respectively, are produced [68]. One such metabolite, betaine, can act as an osmolyte and is a methyl donor. Dietary betaine supplements have potential beneficial effects on human liver disease [69]. Ergotamine is an effective and valuable substance in the treatment of acute migraine attacks [70]. We also identified nine anthocyanin metabolites (Appendix A). These are water-soluble flavonoids responsible for the color of plants. At 70–90 °C and 300 MPa, blueberry puree showed elevated quantities of petunidin-3-*O*-arabinoside, malvidin-3-*O*-galactoside, and malvidin-3-*O*-glucoside [71]. Hence, the release of anthocyanin pigments from epidermal cells resulted in better color retention of the fruit puree [71]. The highest concentrations of the three anthocyanins cyanidin 3-*O*-galactoside, petunidin 3-*O*-glucoside, and cyanidin 3-*O*-glucoside were found in the leaves of the novel purple-leaf tea cultivar Zikui (*Camellia sinensis* cv. Zikui) at 15 days after harvest, which participated in promoting the color of purple leaves [72]. These three anthocyanins were found to be up-regulated in dark green bitter gourd (Appendix A), suggesting their involvement in the pigment accumulation process of the peel color of bitter gourd. Anthocyanins also demonstrate beneficial effects in diseases related to oxidative stress, such as cardiovascular and neurodegenerative diseases [73].

## 4. Materials and Methods

### 4.1. Plant Cultivars

Seven cultivars of bitter gourd, including Pingguo (PG, Taiwan Xinyun Industrial Company, Tainan, Taiwan), Jinlingzi (JLZ, Xinbang Town, Songjiang District, Shanghai, China), Baifumei (BFM, Guangzhou Wangyou Seed Research and Development Co., Ltd., Guangzhou, China), Riben (RB, Kazuki City, Chiba Prefecture, Japan), Changbai (CB, Hebei Cangzhou Jinke Lifeng Seedling Co., Ltd., Hebei, China), Lvjian (LJ, Shouguang City, Weifang, China), and Lvbaoshi (LBS, Vegetable Research Institute of Guangdong Academy of Agricultural Sciences, Guangzhou, China), were collected from the Shanghai Academy of Agricultural Sciences (SAAS). In August 2022, they were planted at Zhuanghang Comprehensive Experimental Station of SAAS (China, 30° N, 121° E). At the commercially full maturity stage, the fruits were collected, washed using sterile water, cut into small pieces, covered in foil paper, and harvested in liquid nitrogen. Samples were taken in triplicate and frozen at a temperature of −80 °C for further evaluation.

### 4.2. Determination of Nutrient Composition Content

The content of ascorbic acid (AsA, Vc) was determined using ASA-2A-W Kit (Suzhou Comin Biotechnology Co. Ltd., Suzhou, China). AsA oxidase can oxidize AsA to dehydroascorbic acid, and hence calculating the AsA concentration is as simple as monitoring the rate of AsA oxidation. The AsA content was presented as μg g^−1^ of the fresh weight of the fruit.

The Coomassie brilliant blue method was implemented to determine the total soluble protein concentration (mg g^−1^ fresh weight of the fruit) [74]. Absorbances were recorded at a wavelength of 595 nm.

The free amino acid content was determined using AA-1-W Kit (Suzhou Comin Biotechnology Co. Ltd., Suzhou, China), which is based on the chromogenic method of ninhydrin solution. The absorbance was gauged at a wavelength of 570 nm.

We used the anthrone-H_2_SO_4_ colorimetric technique to determine the cellulose content. Briefly, the fruit samples were first incubated in 80% ethanol at 65 °C overnight. Subsequently, tissues were exchanged with acetone, and the dried cell wall material was ground to fine powder [75].

Total acid content was determined using the acid-base neutralization titration method described previously [76].

### 4.3. Quantification of Organic Acid Content

The concentration of organic acids was analyzed using an HPLC system (Thermo Vanquish) as per the method described previously [77] with modifications. The bitter gourd sample (0.1 g) was dissolved in distilled water (10 mL), sonicated (45 min), and centrifuged (5000 rpm, 10 min). A column (Agilent InfinityLab Poroshell 120 EC-C18; 4.6 × 50 mm, 2.7 μm) was used to separate the supernatant at 30 °C; the mobile phase was a 95:5 mixture of water and acetonitrile (both in 0.1% formic acid, HPLC grade). The retention time and calibration peaks were determined using standard solutions of oxalic acid, succinic acid, citric acid, and malic acid (Sigma-Aldrich, St. Louis, MO, USA) prepared in pure deionized water.

### 4.4. Total Flavonoid, Triterpenoid, and Phenolic Content Quantification

The Folin–Ciocalteu test was employed [23] to calculate the TPC of bitter gourd. A 1 mL extract of the sample was combined with Folin–Ciocalteu reagent (1 mL) during the experiment. For 2 min, this mixture was incubated (in the dark) at 25 °C. Then, after thorough mixing, 3 mL of a 20% Na_2_CO_3_ solution was added, and the whole mixture was incubated (in the dark) for another 30 min. The microplate reader was used to measure the absorbance of the solution at 717 nm, and the TPC was written as mg of gallic acid equivalents per gram of fresh bitter gourd.

The aluminum chloride colorimetric technique was used to determine the TFC [78]. Here, the extract of the sample (108 μL) was mixed with 5% NaNO_2_ (6 μL) and incubated at 25 °C for 6 min. Subsequently, 10% AlCl_3_ (6 μL) was added, followed by the addition of NaOH (1 mol/L; 80 μL) after 15 min at 25 °C. The microplate reader was used to measure the absorbance of the solution at 510 nm, and the TFC was recorded as the number of milligrams of quercetin equivalents per gram of fresh bitter gourd.

To determine the TTC, a previously established approach was used [79] with slight modifications. The bitter gourd sample (0.1 g) was first ground into powder. Subsequently, the powder was supplied with ethyl acetate (15 mL), sonicated for 30 min, and filtered to obtain the sample extract. This extract was then treated with perchloric acid (1.2 mL) along with a 500 μL solution of vanillin–glacial acetic acid (5%). Then, after 15 min, the liquid was chilled on ice after being incubated at 70 °C. The resulting mixture (40 μL) was then incubated at room temperature after acetic acid (160 μL) was added. At 555 nm, the absorbances were determined.

### 4.5. Determination of Antioxidative Activity

The DPPH assay was performed as described previously [80]. After incubating at 25 °C for 30 min in the dark, the sample solution (10 µL) was combined with 0.066 mM DPPH solution (39 µL). Using a microplate reader, the absorbance was determined to be 515 nm. In this assay, Vc (10 mg mL^−1^) acted as a positive control.

The FRAP assay was conducted as described previously [81]. In this assay, freshly prepared FRAP reagent (10:1:1 [*v*/*v*/*v*] ratio of acetate buffer [300 mM], tripyridyltriazine (TPTZ; 10 mM) solution, and FeCl3 [20 mM]) was mixed with the sample solution and incubated at 37 °C for 20 min in the dark. Subsequently, we combined 5 µL of sample solution, 25 µL of distilled water, and 175 µL of FRAP solution for 10 min at 37 °C. At 595 nm, the absorbance was determined.

In the ABTS assay [82], the ABTS stock solution (prepared by mixing ABTS [7 mM; 10 mL] and potassium persulfate [2.45 mM] diluted with ethanol and water [5 mL]) was incubated at 25 °C in the dark overnight. This solution had an absorbance of 0.70 ± 0.05 at 750 nm. To evaluate samples, the sample solution (10 µL) was mixed with diluted ABTS solution (190 µL). After 10 min of incubation, the absorbances were measured at 734 nm.

### 4.6. UPLC-MS/MS and Secondary Metabolomics Analysis

All bitter gourd samples were freeze-dried in advance of UPLC-MS/MS analysis using a vacuum freeze-dryer (Scientz-100F, Scientz, Ningbo, China). Freeze-dried samples were subsequently crushed using a mixer mill (MM 400, Retsch, Haan, Germany) fitted with a zirconia bead at 30 Hz for 1.5 min. After obtaining the powder (100 mg), 70% methanol (1.2 mL) was added, and the mixture was stirred for 30 s at 30 min intervals (six times in all) before being kept overnight at 4 °C. After centrifuging at 12,000 rpm for 10 min, the methanol extracts were filtered using a microporous filter membrane with a pore size of 0.22 μm (SCAA-104, ANPEL, Shanghai, China, http://www.anpel.com.cn/, accessed on 30 October 2022).

Liquid chromatography–electrospray ionization-MS (LC-ESI-MS/MS) system (Shim-pack UFLC SHIMADZU CBM30A, Kyoto, Japan) and MS (QTRAP^®^ 4500+ System, AB Sciex, Framingham, MA, USA) equipped with a C18 column (ACQUITY UPLC HSS T3, 1.8 µm, 2.1 mm × 100 mm, Waters, Milford, MA, USA) were used in this experiment. During the analysis, the column temperature, flow rate, and injection volume were 40 °C, 0.4 mL/min, and 2 μL, respectively. Furthermore, the solvent system constituted water and acetonitrile (both in 0.1% formic acid). The following gradient program was used during the experiment: 95:5, 5:95, 5:95, 95:5, and 95:5 (*v*/*v*) at 0 min, 10.0 min, 11.0 min, 11.1 min, and 15.0 min, respectively.

Positive as well as negative ion modes were used during the MS analysis. The following were the ESI source operation parameters: (1) an ion source temperature of 550 °C; (2) ion spray voltage 5500 V (positive ion mode)/−4500 V (negative ion mode); (3) ion source gas I 50 psi, gas II 60 psi, the curtain gas 25 psi; and (4) high collision-activated dissociation. Multiple reaction monitoring experiments were employed to acquire LIT and triple quadrupole (QQQ) scans; during this analysis, the medium contained the collision gas (nitrogen). During QQQ scanning, each ion pair was scanned and detected based on optimized clustering potential and collision energy. All the data were controlled and processed using Analyst 1.6.3 software (AB Sciex, Framingham, MA, USA).

In context to secondary metabolome data of bitter gourds, the results of the orthogonal projections to latent structure discriminant analysis were used to extract VIP values, which represents the intensity of the influence of the inter-group differences of corresponding metabolites on the classification and discrimination of each group of samples in the model. The relative content of each compound was calculated using the internal standard normalization method. Significantly changed metabolites between groups were determined using VIP value ≥ 1 and FC ≥ 2 or ≤0.5.

### 4.7. KEGG Annotation and Enrichment Analysis

To elucidate the overall metabolic (between samples) and metabolome (between groups) differences, the principal component and Pearson’s correlation analyses were used [83]; statistical function Prcomp of R software (version 4.2.3, https://www.r-project.org/) was employed. Furthermore, the pheatmap function of R software was used to conduct the hierarchical cluster analysis (HCA) of identified metabolites, and Microsoft Excel 2019 was used to construct volcano plots.

Since JLZ is a wild variety [20], it was used as the control in this study. The differential metabolites between different groups were annotated using the KEGG compound database (http://www.kegg.jp/kegg/compound/, accessed on 30 December 2022) and mapped to the KEGG pathway database (http://www.kegg.jp/kegg/pathway.html, accessed on 30 December 2022) for significant enrichment analysis and the identification of major enriched pathways [84]. KEGG pathway enrichment analysis of differential metabolites was performed using metabolite sets enrichment analysis (MESA), which was determined using *p*-values of the hypergeometric test.

### 4.8. Statistical Analysis

The SPSS 25 software was used to perform all the statistical analysis. All the experiments were conducted in triplicate. Results are presented as means ± standard deviations (SDs). Furthermore, Student’s two-tailed t-test was used to determine significant differences between the two groups. The statistical significance of the results was adjudged on the basis of a *p* value of <0.05.

## 5. Conclusions

In conclusion, this study displayed that the LJ cultivar of bitter gourd had the highest TPC, TFC, and TTC content, demonstrating the highest level of antioxidant activity. Other cultivars (LBS and CB) also showed varying levels of TPC, TFC, and TTC along with antioxidant activities. This study identified 370 secondary metabolites, which included 90, 76, and 56 flavonoids, phenolic acids, and alkaloids, respectively. These compounds were profiled using extensively targeted metabolomics in seven cultivars of bitter gourd. Furthermore, 27 compounds were significantly related to the antioxidative capacity of bitter gourds. Compared with the JLZ cultivar, the PG cultivar had the most significant number of significantly different metabolites, whereas other cultivars had similar amounts of significantly different metabolites, albeit their change patterns were different. All cultivars of bitter gourd were found to be enriched in flavonoid, folate, flavone, flavonol, isoflavonoid, and phenylpropanoid biosynthesis pathways; however, the composition and pattern were diverse among different cultivars. The data presented in this study greatly enrich our understanding of different metabolites present in the plants belonging to the Cucurbitaceae family. According to the current study, different commercial bitter gourd varieties can also be used to produce different high-value products, such as antioxidative compounds, functional components, health products, and nutrition products. The study also examined the differences in biological activity potential and metabolic changes in bitter gourd cultivars in different regions and different peel colors. This study contributes to our knowledge of the metabolic pathways in bitter gourd and helps in the assessment of metabolic quality.

## Figures and Tables

**Figure 1 ijms-24-14495-f001:**
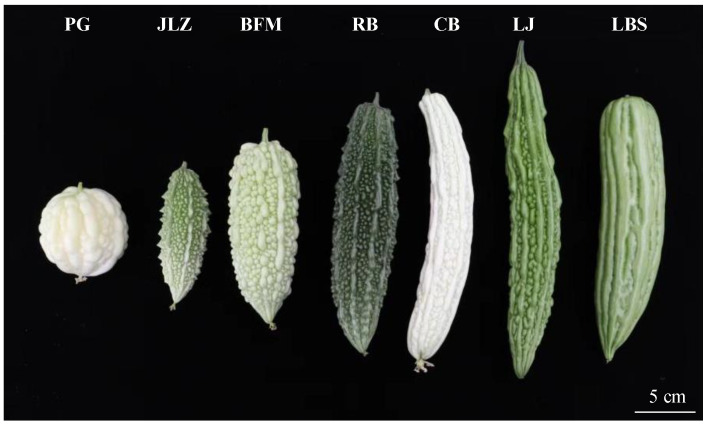
Morphology of fruit harvested 18 days after simultaneous pollination.

**Figure 2 ijms-24-14495-f002:**
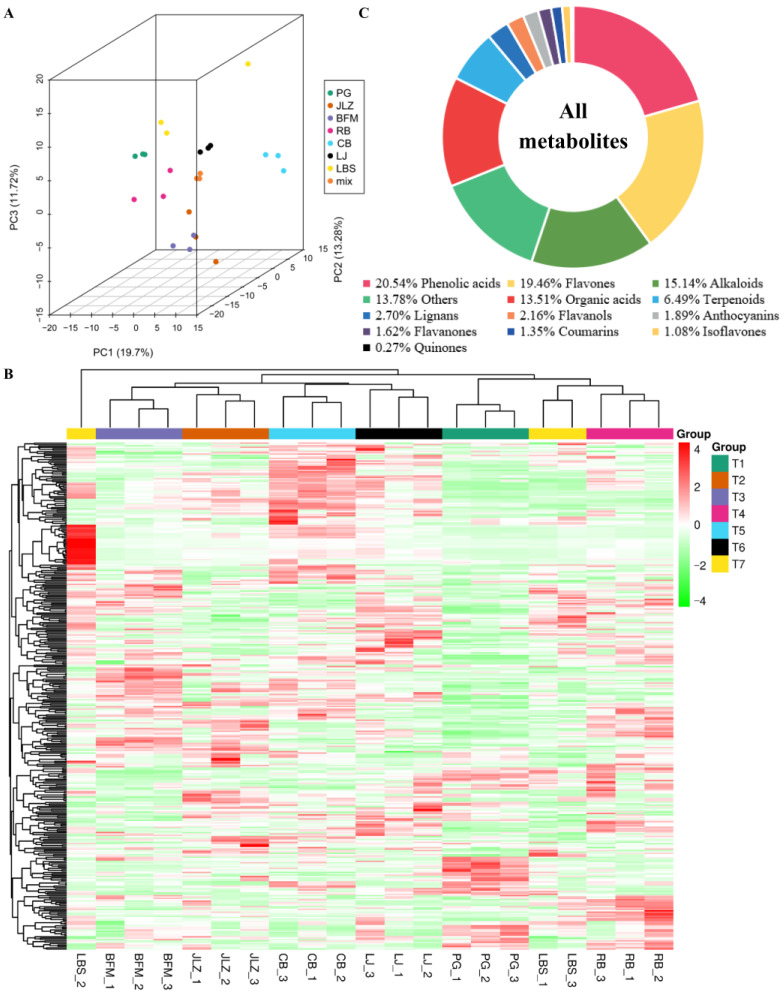
Overview of the analysis conducted to identify metabolites in seven different cultivars of bitter gourd. (**A**) Principal component analysis of the relative presence of metabolites in PG, JLZ, BFM, RB, CB, LJ, and LBS. (**B**) Cluster heatmap for metabolites content in different cultivars of bitter gourd. (**C**) Classification of secondary metabolites in different cultivars of bitter gourd.

**Figure 3 ijms-24-14495-f003:**
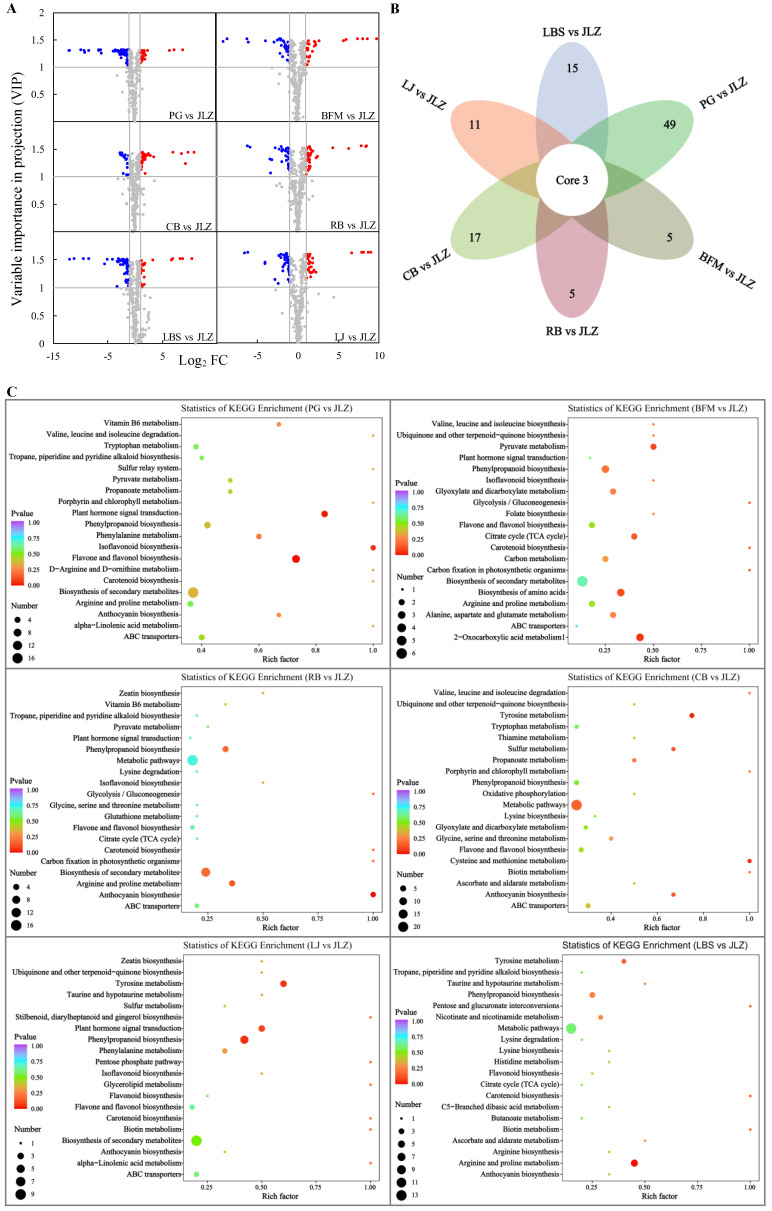
Differentially accumulated metabolites, Venn diagram, and analysis of different pathways in different cultivars of bitter gourd. (**A**) The volcano plot shows the differences in the expression levels of metabolites among different samples. Red, blue, and gray dots indicate up-regulated, down-regulated, and no significant difference in metabolite expression, respectively. (**B**) The Venn diagram shows the overlapping and cultivar-specific differences in the metabolites’ accumulation. (**C**) Enrichment of the KEGG pathway for differentially accumulated metabolites among different cultivars.

**Figure 4 ijms-24-14495-f004:**
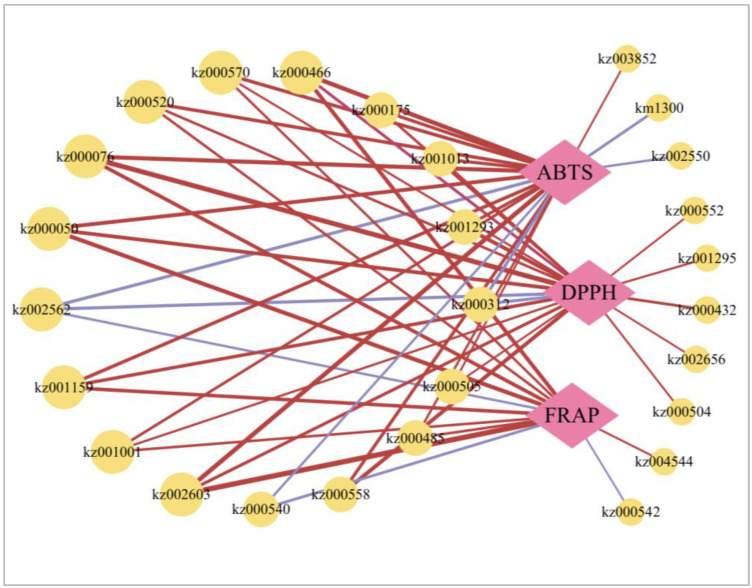
Network diagram. Pink diamonds and yellow circles indicate different antioxidative capacities and metabolites, respectively (the number represents Ko_ID). Correlation is represented by the line connecting the circle and the diamond, and the thicker lines indicate more strong correlation. Positive and negative correlations are represented in red and blue, respectively (r ≥ 0.7 and *p* < 0.05).

**Table 1 ijms-24-14495-t001:** Comparison of nutrient contents in pulp of seven bitter gourd cultivars.

Cultivar	Vc Content (μg/g FW)	Soluble Protein Content (mg/g FW)	Amino Acid Content (mg/g FW)	Cellulose Content (mg/g DW)	Total Acid Content (%)
PG	117.28 ± 3.58 ^a^	4.70 ± 0.34 ^d^	1.05 ± 0.04 ^e^	298.79 ± 15.28 ^bc^	0.41 ± 0.01 ^cd^
JLZ	121.55 ± 11.88 ^a^	12.83 ± 0.51 ^a^	1.64 ± 0.02 ^c^	284.81 ± 7.45 ^c^	0.28 ± 0.00 ^e^
BFM	56.65 ± 1.59 ^b^	5.13 ± 0.06 ^d^	2.03 ± 0.03 ^a^	399.46 ± 5.52 ^a^	0.40 ± 0.01 ^d^
RB	125.83 ± 6.08 ^a^	6.07 ± 0.34 ^c^	0.52 ± 0.03 ^f^	274.94 ± 2.83 ^c^	0.47 ± 0.00 ^b^
CB	126.57 ± 9.16 ^a^	11.52 ± 0.17 ^b^	1.16 ± 0.02 ^d^	372.33 ± 2.96 ^a^	0.40 ± 0.00 ^cd^
LJ	65.28 ± 5.09 ^b^	6.20 ± 0.02 ^c^	1.86 ± 0.05 ^b^	318.35 ± 21.41 ^b^	0.52 ± 0.02 ^a^
LBS	63.07 ± 0.78 ^b^	13.30 ± 0.57 ^a^	2.03 ± 0.05 ^a^	287.31 ± 19.39 ^c^	0.42 ± 0.00 ^c^

Means in columns followed by the same letters are not significantly different.

**Table 2 ijms-24-14495-t002:** The organic acid of seven bitter gourd cultivars.

Cultivar	Oxalic Acid (mg/kg)	Succinic Acid (mg/kg)	Malic Acid (mg/kg)	Citric Acids (mg/kg)
PG	206.05 ± 13.34 ^de^	44.19 ± 2.12 ^b^	853.54 ± 50.04 ^d^	41.02 ± 2.07 ^d^
JLZ	341.86 ± 31.26 ^ab^	26.68 ± 0.73 ^c^	2214.07 ± 123.42 ^a^	186.20 ± 18.59 ^a^
BFM	221.07 ± 14.99 ^de^	26.50 ± 1.05 ^c^	1092.73 ± 81.41 ^c^	107.19 ± 11.44 ^b^
RB	255.47 ± 25.31 ^cd^	30.30 ± 3.28 ^c^	800.29 ± 81.34 ^d^	43.84 ± 2.27 ^d^
CB	302.29 ± 25.27 ^bc^	63.10 ± 2.18 ^a^	2121.76 ± 56.72 ^a^	108.38 ± 7.18 ^b^
LJ	365.75 ± 67.11 ^a^	20.47 ± 2.11 ^d^	1343.16 ± 85.76 ^b^	68.90 ± 11.12 ^c^
LBS	164.22 ± 6.11 ^e^	16.79 ± 2.36 ^d^	594.00 ± 76.59 ^e^	37.78 ± 3.65 ^d^

Means in columns followed by the same letters are not significantly different.

**Table 3 ijms-24-14495-t003:** The total phenolic content (TPC), total flavonoid content (TFC), total triterpenoids content (TTC), and antioxidant activities of seven bitter gourd samples.

Cultivar	TPC (μg/g)	TFC (mg/100 g)	TTC (mg/g)	DPPH (%)	ABTS (μmol/g FW)	FRAP (μmol/g FW)
PG	17.02 ± 0.64 ^d^	130.33 ± 9.58 ^e^	1.23 ± 0.06 ^e^	0.58 ± 0.02 ^e^	10.11 ± 0.19 ^g^	9.64 ± 0.62 ^d^
JLZ	17.60 ± 1.14 ^d^	135.85 ± 7.26 ^de^	1.24 ± 0.11 ^e^	0.62 ± 0.02 ^e^	10.50 ± 0.30 ^f^	9.83 ± 0.66 ^cd^
BFM	20.11 ± 1.02 ^c^	142.58 ± 16.26 ^cde^	1.33 ± 0.02 ^de^	0.68 ± 0.01 ^d^	11.11 ± 0.18 ^e^	10.14 ± 0.39 ^cd^
RB	21.04 ± 0.75 ^c^	150.94 ± 8.33 ^bcd^	1.45 ± 0.08 ^cd^	0.75 ± 0.03 ^c^	11.80 ± 0.25 ^d^	10.84 ± 0.48 ^c^
CB	24.04 ± 0.99 ^b^	160.24 ± 7.45 ^bc^	1.72 ± 0.06 ^a^	0.84 ± 0.02 ^a^	12.60 ± 0.10 ^c^	12.84 ± 0.63 ^b^
LJ	29.05 ± 1.23 ^a^	180.54 ± 10.46 ^a^	1.59 ± 0.08 ^ab^	0.81 ± 0.05 ^ab^	13.48 ± 0.03 ^a^	13.93 ± 0.53 ^a^
LBS	25.26 ± 1.20 ^b^	170.59 ± 12.73 ^ab^	1.49 ± 0.06 ^bc^	0.77 ± 0.03 ^bc^	13.00 ± 0.12 ^b^	13.33 ± 0.63 ^ab^

Means in columns followed by the same letters are not significantly different.

**Table 4 ijms-24-14495-t004:** Correlation analysis of TPC, TFC, TAC, and antioxidant capacity.

		TPC	TFC	TTC	DPPH	ABTS	FRAP		TPC	TFC	TTC	DPPH	ABTS	FRAP
TPC	PG	1	−0.997 **	−0.997 **	−0.507 **	−0.983 **	0.999 **	JLZ	1	−0.989 **	−0.551 **	−0.925 **	−0.991 **	−0.976 **
TFC			1	1	0.573 **	0.995 **	−0.995 **			1	0.671 **	0.972 **	1 **	0.998 **
TTC				1	0.572 **	0.994 **	−0.996 **				1	0.826 **	0.657 **	0.720 **
DPPH					1	0.655 **	−0.492 **					1	0.968 **	0.986 **
ABTS						1	−0.980						1	0.996
FRAP							1							1
TPC	BFM	1	−0.976 **	0.133 **	0.908 **	−0.459 **	0.581 **	RB	1	0.181 **	0.597 **	−0.508 **	0.418 **	0.410 **
TFC			1	−0.345 **	−0.796 **	0.640 **	−0.744 **			1	−0.681 **	0.755 **	0.969 **	−0.823 **
TTC				1	−0.294 **	−0.942 **	0.884 **				1	−0.994 **	−0.479 **	0.976 **
DPPH					1	−0.045 **	0.187 **					1	0.570 **	−0.994 **
ABTS						1	−0.990 *						1	−0.657 *
FRAP							1							1
TPC	CB	1	−0.467 **	−0.558 **	−0.960 **	1 **	0.959 **	LJ	1	−0.993 **	0.340 **	−0.516 **	0.878 **	−0.960 **
TFC			1	−0.473 **	0.695 **	−0.469 **	−0.698 **			1	−0.446 **	0.414 **	−0.817 **	0.986 **
TTC				1	0.304 **	−0.556 **	−0.301 **				1	0.630 **	−0.152 **	−0.591 **
DPPH					1	−0.961 **	−1 **					1	−0.863 **	0.254 **
ABTS						1	0.960						1	−0.708
FRAP							1							1
TPC	LBS	1	−0.200 **	0.975 **	−0.382 **	0.407 **	0.302 **							
TFC			1	−0.413 **	−0.829 **	−0.976 **	−0.994 **							
TTC				1	−0.167 **	0.600 **	0.506 **							
DPPH					1	0.689 **	0.765 **							
ABTS						1	0.994							
FRAP							1							

Red means positive correlations, blue means negative correlations; ** Correlation is significant at the 0.01 level, and * correlation is significant at the 0.05 level (2-tailed).

## Data Availability

The data presented in this study are available on request from the corresponding author.

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
