# Peer review of "Integrated Secondary Metabolomic and Antioxidant Ability Analysis Reveals the Accumulation Patterns of Metabolites in Momordica charantia L. of Different Cultivars"

_ijms, 2023, doi:10.3390/ijms241914495_

Round 1
Reviewer 1 Report
Reviewer comments and suggestions
The authors in this study systematically quantified the nutrient contents, and the total content of phenolic acids (TPC), flavonoids (TFC), and triterpenoids (TTC) in seven different cultivars of bitter gourd.
They also estimated the organic acid content and antioxidative capacity of different cultivars of bitter gourd. In the metabolomics experiment, 370 secondary metabolites were identified in seven cultivars of bitter gourd; flavonoids and phenolic acids were significantly more.
The study concluded by saying that the outcomes may suggest bitter gourd consist of plethora of bioactive compounds, and hence bitter gourd may potentially be applied in developing novel molecules of medicinal importance.
Overall, the manuscript was well written. However, a few major concerns/comments needed to be explained/modified.
- Line 37-39 The authors could add up few information here
- Line 45-46 I think one reference is not suitable for these all, please provide more
- Line 68-69 I think these lines are not important enough to be here in the introduction section
- Line 266-267 Please mention a specific paragraph for the discussion before mentioning the subsections
- Line 324-328 Did the authors check these metabolite ( please mention the table or figure)?
- Line 353 What does the line indicate, Please explain it well
- Section 4.7 The section should be well discussed here
- All references should be modified based on MDPI.
Reviewer 2 Report
Comments
In general, the submitted manuscript to IJMS evaluated the amount of different metabolites in bitter gourd. I trust that the paper has the publication potential, however it should be improved in various aspects, as mentioned in the following comments:
What is the novelty of this study? These content analyses can be done by any simplified laboratory!
Line 19: “experiment” should be replaced with “analysis”
The results should be improved by providing the difference (in percentage or fold-change) among cultivars for different metabolites and antioxidants.
Why only these enzymes were selected to analyze?
It would be very nice if the authors can do qPCR analysis for the most important genes for this study.
Recommendation: Minor Revision
Minor editing of English language required
